# Individual and Cumulative Health and Lifestyle Risk Factors for Depressive Symptoms in Older Adults: Evidence from NHANES

**DOI:** 10.3390/geriatrics11010005

**Published:** 2026-01-01

**Authors:** Chaowalit Srisoem, Mia Haddad, Jittima Panyasarawut, Ling Shi

**Affiliations:** 1Department of Nursing, College of Nursing and Health Sciences, University of Massachusetts Boston, Boston, MA 02125, USA; chaowalit.srisoem001@umb.edu (C.S.); mia.picillo001@umb.edu (M.H.); j.panyasarawut001@umb.edu (J.P.); 2Department of Psychiatric and Mental Health Nursing, Boromarajonani College of Nursing Sunpasitthiprasong, Faculty of Nursing, Praboromarajchanok Institute, Ubon Ratchathani 34000, Thailand; 3Department of Adult and Geriatric Medicine, Massachusetts General Hospital, Boston, MA 02114, USA; 4Department of Fundamental Nursing, Faculty of Nursing, Mahidol University, Nakhon Pathom 73170, Thailand

**Keywords:** older adults, depressive symptoms, physical health, lifestyle

## Abstract

Background: Depression in older adults is a multifactorial condition influenced by health status, functional capacity, and lifestyle factors. This study aimed to investigate the individual and combined associations of these factors with late-life depression. Methods: Using data from the National Health and Nutrition Examination Survey (NHANES), this study evaluated the associations of general health, chronic conditions, functioning, and lifestyle behaviors (including physical activity, sleep, diet quality, smoking, and alcohol use) with depressive symptoms among U.S. adults 65 years and older. Weighted logistic regression models, accounting for the complex survey design of NHANES, were used to examine the factors both individually and in combination. Results: Depressive symptoms were more prevalent among individuals with poor self-rated health, physical and cognitive functional limitations, hypertension, obesity, current smoking, physical inactivity, and alcohol abstinence. A clear cumulative risk gradient was observed with increasing numbers of risk factors: older adults with six or more risk factors had at least 20-fold higher likelihood of depressive symptoms compared with those with one or no risk factors. Conclusions: These findings highlight the interdependent influences of health, function, and lifestyle on late-life depressive symptoms and underscore the need for integrative prevention and intervention strategies to promote mental well-being in aging populations.

## 1. Introduction

Depression is a prevalent and disabling condition among older adults, affecting approximately 10% to 20% of this population globally [1]. Its consequences extend beyond emotional well-being, contributing to diminished quality of life, increased risk of physical and cognitive impairment, functional decline, social isolation, and greater healthcare utilization [2,3]. With the rapid aging of populations worldwide, late-life depression has become a pressing public health concern.

A growing body of literature has explored factors associated with depression in older adults. Poor general health and the presence of multiple chronic conditions including hypertension, diabetes, cardiovascular disease, and obesity are consistently linked to late-life depression [4,5]. These conditions may contribute to depression through inflammatory pathways, reduced mobility, or increased treatment burden, which collectively compromise well-being.

Functional impairments are another key contributor. Limitations in physical functioning such as difficulty walking, climbing stairs, or performing daily self-care have consistently emerged as strong correlates of depression in older adults [2]. Such limitations reduce autonomy, increase dependence on others, and may foster social withdrawal, exacerbating depressive symptoms. Cognitive decline, even at a mild level characterized by memory lapses or confusion, is also associated with depressive symptoms [1]. The relationship between cognition and depression is often bidirectional: depression can accelerate cognitive decline, while cognitive impairment can diminish coping abilities and increase vulnerability to depressive symptoms.

Lifestyle behaviors are increasingly recognized as modifiable determinants of depression in later life. Regular physical activity is associated with better mood, enhanced cognitive performance, and improved quality of life [3]. Conversely, physical inactivity is linked to both depression and increased cardiovascular risk [6,7]. Sleep patterns also play an important role. Both short and long sleep durations are associated with worse mental health outcomes, while recommended sleep duration (7–9 h) is protective [8]. Dietary quality has been linked to mental health, with adherence to healthy dietary patterns, such as those captured by the Healthy Eating Index (HEI-2015), associated with lower depression risk [9]. Smoking and heavy alcohol use further compound mental health risks, while moderate alcohol consumption may have a more complex relationship with depressive symptoms [4].

While these factors such as chronic disease, functional status, and lifestyle behaviors have been studied individually, few studies have examined how the accumulation of these factors influences depression among older adults. Existing research often fails to capture the multifactorial and interdependent nature of late-life depression. Growing evidence highlights that late-life depression is inherently multifaceted, arising from the mutual influence among biological, psychological, and social determinants. Rather than functioning independently, these factors interact collectively to shape vulnerability, symptom severity, and clinical course in older adults [10,11]. Understanding these combined effects is critical for informing integrative prevention and intervention strategies.

To address this gap, the current study utilized data from the National Health and Nutrition Examination Survey (NHANES), a nationally representative survey of the United States (U.S.) population to examine the independent and combined associations of general health, physical and cognitive function, and lifestyle behaviors with depressive symptoms among older adults. This comprehensive approach may provide a more nuanced understanding of late-life depression and inform multifaceted strategies to improve mental health outcomes in aging populations.

## 2. Materials and Methods

The study utilized NHANES data, a nationally representative survey conducted every two years since 1999 by the National Center for Health Statistics under the Centers for Disease Control and Prevention (CDC). We used data from three cycles collected over 6 years, from 2013 to 2018. Later NHANES cycles did not include certain key variables, such as measures of diet quality and functioning, and were therefore excluded from the analysis. NHANES employs a multistage probability sampling design to assess the health and nutritional status of the civilian, noninstitutionalized U.S. population [12]. NHANES data were collected using interviewer-administered questionnaires and standardized physical examinations performed at mobile examination centers (MECs). Participants who consent to clinical assessments undergo anthropometric measurements, blood pressure evaluations, and laboratory tests, including the collection of blood and urine samples. Additional details on NHANES methodologies can be found in official publications [12]. This study included 3942 participants aged 65 years and above who were well-cognizant of answering the Patient Health Questionnaire-9 (PHQ-9) and were eligible to participate in the study.

### 2.1. Variables/Measures

#### 2.1.1. Depressive Symptoms

Depressive symptoms were assessed using the PHQ-9. The PHQ-9 is a well-validated instrument consisting of nine items that measure the presence and severity of depressive symptoms experienced during the preceding two weeks. Each item is scored on a 4-point scale ranging from 0 (no occurrence of the symptom in the past 2 weeks), 1 (occasional presence on several days), 2 (frequent occurrence, i.e., more than half the days), to 3 (nearly every day), with total scores ranging from 0 to 27. Higher scores indicate greater symptom severity. For this study, presence of depressive symptoms was dichotomized using a standard cutoff score ≥10 indicating the presence of depressive symptoms [13]. This cutoff value was chosen for its high sensitivity (88%) and specificity (88%) for major depression [13].

#### 2.1.2. General Health

Perceived health status was measured by asking participants, “How would you rate your overall health?” Responses were classified into three categories: (1) excellent or very good, (2) good, and (3) fair or poor. This simple self-assessment provides a useful indicator of individuals’ general well-being and is acceptable for use in NHANES research [14].

Hypertension was defined as meeting one of the following criteria: (1) a systolic blood pressure (SBP) ≥ 140 mmHg, (2) diastolic blood pressure (DBP) ≥ 90 mmHg, or (3) self-reported hypertension diagnosis, or the utilization of prescribed antihypertensive medication [5,7]. Blood pressure measurements were taken according to the American Heart Association (AHA) standardized protocol. Certified examiners conducted measurements in the MECs using a mercury sphygmomanometer and appropriately sized cuffs [15]. Prior to measurement, participants were instructed to rest quietly in a seated position for five minutes, and their maximum inflation level was determined [16]. Three consecutive blood pressure readings were obtained at 30 s intervals. If a reading was interrupted or incomplete, a fourth attempt was made. The means of the three valid measurements determined blood pressure levels [15,16].

Body weight status was categorized into four groups based on BMI: underweight (BMI < 18.5 kg/m^2^), normal weight (BMI 18.5–24.9 kg/m^2^), overweight (BMI 25.0–29.9 kg/m^2^), and obese (BMI ≥30.0 kg/m^2^) [17]. BMI was calculated as weight (kg) divided by height squared (m^2^). Height and weight measurements were obtained during the NHANES examination using standardized protocols and calibrated instruments.

#### 2.1.3. Functioning Status

Subjective cognitive decline question (SDQ) was assessed using a single interview question, which evaluated the experience of worsening memory loss or confusion [18]. Physical functioning was assessed using 14 activities representing physical function and participation restriction, aligned with the International Classification of Functioning, Disability, and Health (ICF) framework [19]. These were grouped into (1) activities of daily living, such as social participation, self-care, mobility within the home, and household tasks, and (2) physical performance activities assessing strength, mobility, and balance. NHANES participants rated their difficulty in performing each activity independently without equipment using a four-point scale (“no difficulty” to “unable to perform”). Nonresponses were treated as missing data. Scores were coded from 1 (no difficulty) to 4 (unable to do), and a mean score was calculated for each participant. Individuals with an average score of 1 were classified as having no limitations, whereas those with a mean score ≥2 were classified as having at least one physical functional activity limitation [20]. 

#### 2.1.4. Lifestyle Factors

Physical activity (PA) data were collected using the Global Physical Activity Questionnaire (GPAQ) [6]. Participants reported their PA over 30 days across leisure-time, occupational, and recreational domains. The GPAQ measures moderate- and vigorous-intensity PA minutes weekly. PA levels were quantified using AHA guidelines by multiplying daily minutes by the number of active days per week for each intensity level [21]. Total PA minutes/weekly were calculated by doubling vigorous PA minutes and adding moderate PA minutes. Participants were classified as “meeting AHA guidelines” if they achieved ≥150 min total of PA per week and not “meeting AHA guidelines” if they achieved <150 min of total PA per week [21].

Smoking status was defined using self-reported data from the smoking-cigarette use questionnaire [12]. Participants were asked whether they had smoked at least 100 cigarettes in their lifetime. Based on their responses, smoking status was categorized into three groups: (1) never smokers (those who had not smoked 100 cigarettes in their lifetime); (2) current smokers (those who had smoked at least 100 cigarettes in their lifetime and currently smoke); and (3) former smokers (those who had smoked at least 100 cigarettes in their lifetime but no longer smoke).

Alcohol consumption was assessed based on the frequency of alcohol use in the past year. Responses were classified into five categories: never, rarely (1–11 times), monthly (1–3 times per month), weekly (1–4 times per week), and often or daily (nearly every day or every day). The categories were collapsed into three groups: no drinking in the past year, occasionally (rarely and monthly), and regular drinking (weekly, often or daily).

Sleep duration was assessed using the NHANES sleep disorders questionnaire. “How much sleep do you usually get at night on weekdays or workdays?” and “How much sleep do you usually get at night on weekends?” The total average sleep duration per night was then calculated in hours and categorized into three categories based on the recommendations from the American Academy of Sleep Medicine and the Life’s Essential 8 [7,8]: (1) short sleep (<7 h per night); (2) recommended sleep (7–9 h per night) as the reference group, and (3) long sleep (>9 h per night).

Diet quality was assessed using the HEI-2015, a validated tool that measures adherence to the Dietary Guidelines for Americans (DGA) [9]. Dietary data were collected through 24 h recall interviews by trained interviewers. The HEI-2015 has 13 dietary components in adequacy and moderation [9]. Adequacy includes total fruits, whole fruits, total vegetables, greens and beans, total protein foods, seafood, and plant proteins (5 points each), plus fatty acids, whole grains, and dairy (10 points each). Moderation comprises refined grains, sodium, added sugars, and saturated fat (10 points each) [9]. The HEI-2015 score ranges from 0 to100, with higher scores indicating better diet quality and greater alignment with the DGA.

#### 2.1.5. Sociodemographic Covariates

Sociodemographic data were collected through the NHANES demographic questionnaire. Covariates included age, gender (male/female), and race/ethnicity, categorized as non-Hispanic White, non-Hispanic Black, Hispanic, and Other/Multiracial. Educational attainment was classified as less than high school, high school graduate or GED, some college or associate degree, and college graduate. Marital status was grouped into three categories: married/living with a partner, never married, and widowed/divorced/separated.

### 2.2. Data Analysis

A complete case analysis approach was used, including only participants with complete data on all study variables. After excluding ineligible participants with missing values, chi-square tests indicated no significant differences between individuals with and without missing data. All analyses were conducted using STATA version 18 (StataCorp LLC, College Station, TX, USA), accounting for the complex, multistage survey design of NHANES. Weighted descriptive statistics (means and proportions) were calculated to summarize participant characteristics by presence of depressive symptoms. Bivariate associations between predictor variables and presence of depressive symptoms were examined using survey-weighted chi-square tests for categorical variables and t-tests for continuous variables.

To assess the independent associations between general health, functional status, lifestyle factors, and presence of depressive symptoms, multivariable logistic regression models were fitted using survey weights. Predictor variables included general health, physical and cognitive functioning, and lifestyle factors. Covariates were selected based on conceptual relevance and prior literature, and included age, gender, race/ethnicity, education level, and marital status. In addition to examining individual factors, we also assessed the cumulative effects of these modifiable factors on presence of depressive symptoms and addressed their potential intercorrelations by examining the number of risk factors as a predictor. Adjusted odds ratios (AORs) with 95% confidence intervals (CIs) were reported. All statistical tests were two-sided, and a *p*-value < 0.05 was considered statistically significant.

## 3. Results

### 3.1. Characteristics of Participants

The analytic sample comprised 3942 older adults (Table 1). The mean age of participants was 72.69 years (SD = 4.96), with a slight majority being female (55.5%). Most participants were White (78.7%), married (61.3%), and had at least some college education (61.0%). Depressive symptoms were more prevalent among women, Hispanic participants, and individuals with less than a high school education (*p* < 0.05).

### 3.2. Bivariate Associations

Weighted bivariate analyses indicated that older adults with presence of depressive symptoms were significantly more likely to report fair or poor self-rated health, physical functioning difficulties, confusion or memory problems, and have hypertension and obesity. They were also more likely to engage in unhealthy behaviors, including current smoking, insufficient physical activity, and not meeting recommended sleep duration, and were more likely to abstain entirely from alcohol. No significant differences were observed between depressive and non-depressive participants in dietary quality (Table 2).

### 3.3. Multivariable Logistic Regression Results

Results from multivariable logistic regression are presented in Table 3. After adjusting for age, gender, race/ethnicity, marital status, and education levels, older adults with fair or poor self-rated health were nearly six times more likely to experience depressive symptoms compared to those reporting very good or excellent health (AOR = 5.90, 95% CI: 2.68–12.97, *p* < 0.001). Physical functioning difficulties were associated with elevated odds of depressive symptoms (AOR = 2.81, 95% CI: 1.08–7.35, *p* = 0.036). Long sleep duration (AOR = 2.25, 95% CI: 1.19–4.26, *p* = 0.017) and hypertension (AOR = 3.08, 95% CI: 1.08–8.82, *p* = 0.038) were also significantly associated with depressive symptoms. Being overweight was associated with reduced odds of depressive symptoms relative to normal weight (AOR = 0.29, 95% CI: 0.09–0.95, *p* = 0.042).

A cumulative risk gradient was observed with increasing numbers of risk factors (Table 4). Compared with those with one or no risk factors, having two or three risk factors was not associated with significantly higher odds of depressive symptoms. In contrast, individuals with four to five risk factors were approximately five times more likely to experience depressive symptoms (AOR = 5.04, 95% CI: 1.35–18.89, *p* = 0.017; AOR = 5.76, 95% CI: 1.48–22.48, *p* = 0.013, respectively). The odds of depressive symptoms increased markedly with six risk factors (AOR = 20.94, 95% CI: 5.56–78.81, *p* < 0.001), and were highest among individuals with seven to nine risk factors, who were nearly 28 times more likely to be depressed compared with the reference group (AOR = 27.89, 95% CI: 8.46–91.95, *p* < 0.001).

## 4. Discussion

In this cross-sectional study of the adult population aged 65 years and above, we found that several general health, functional and modifiable lifestyle factors were significantly associated with presence of depressive symptoms. Specifically, perceived good health, no physical difficulty, absence of obesity, absence of hypertension, regular sleep duration, nonsmoking status, regular alcohol consumption, and meeting physical activity guidelines were all linked to reduced likelihood of depressive symptoms. These findings highlight the multifactorial nature of late-life depression and underscore the importance of promoting a holistic approach to physical and mental well-being among older adults.

Our findings that perceived good health and absence of hypertension and obesity were negatively associated with depressive symptoms align with previous studies that physical health is an important factor for psychological health in older adults. Chronic conditions such as hypertension and obesity may contribute to depression through biological mechanisms, including systemic inflammation and vascular changes that alter brain structure and function [5]. They also increase functional limitations, pain, and fatigue, which reduce quality of life and increase vulnerability to depressive symptoms [1]. Conversely, better perceived health and fewer chronic illnesses support autonomy, independence, and social participation, thus protecting against depression. Functional impairments further compound these effects, as difficulty in mobility or self-care often leads to social isolation, reduced self-efficacy, and diminished purpose, all of which are strongly associated with depression [2]. Although cognitive impairment was strongly associated with depressive symptoms in univariate analysis, with 36.4% of participants with depressive symptoms versus 9.2% of those without depressive symptoms having confusion/memory problems, the association was no longer statistically significant in multivariable models (OR = 1.61, 95% CI: 0.71–3.62, *p* > 0.05). This may reflect collinearity with other factors such as physical functional status and chronic disease. Nevertheless, the elevated likelihood of depression among those experiencing cognitive problems suggests a potentially meaningful association that warrants further investigation.

Sleep duration was also identified as another key factor, with regular sleep associated with lower risk of depressive symptoms. Sleep disturbances worsen depressive symptoms through disruptions in circadian rhythms, dysregulation of neurotransmitters such as serotonin, and impaired emotional regulation [8]. Both short and long sleep durations have been linked to worse mental health outcomes; short sleep can lead to fatigue, irritability, and cognitive decline, while excessively long sleep may reflect underlying chronic illness or poor health behaviors that coexist with depression. Thus, maintaining adequate and consistent sleep duration may represent a critical target for promoting mental well-being among older adults.

Our results also indicated that nonsmokers had a lower prevalence of depressive symptoms, consistent with research demonstrating the bidirectional relationship between smoking and depressive symptoms. Individuals with depression may be more likely to smoke as a coping mechanism for stress or low mood. In contrast, nicotine dependence and smoking-related biological changes can increase vulnerability to depression through alterations in dopamine and serotonin pathways and by worsening physical health [4]. This highlights the importance of smoking cessation programs that integrate mental health support, especially for older populations where comorbidities increase risks. The loss of statistical significance of smoking in the multivariable model is likely due to collinearity with other factors, such as alcohol consumption and chronic conditions.

Interestingly, regular alcohol drinking was associated with lower risk of depressive symptoms. Moderate alcohol use may facilitate social interaction, relaxation, and stress reduction, which are protective for mental health. However, heavy or harmful drinking patterns have the opposite effect, increasing depression risk and compounding chronic health issues [4]. These findings highlight the complex relationship between alcohol use and mental health, emphasizing the need for cautious interpretation and public health messaging. Further investigation is needed to fully understand this relationship and avoid misinterpretation.

Meeting physical activity guidelines was inversely associated with depressive symptoms, which is consistent with the well-established evidence that regular physical activity improves mood through biological and psychosocial mechanisms. Exercise releases endorphins, improves neuroplasticity, and reduces inflammation, all of which are beneficial for mental health [3]. Beyond biological mechanisms, physical activity provides opportunities for social engagement, structure, and purpose, which further contribute to resilience against depression. In older adults, maintaining mobility and physical function through activity may also delay disability, promote independence and improve overall quality of life.

Although previous studies have reported an inverse association between HEI-2015 and depression [22], our study did not observe a statistically significant relationship in older adults. This discrepancy may reflect age-related differences in dietary patterns or increased measurement error in dietary recall in older adults. Future work using longitudinal dietary measures and alternative HEI categorizations may clarify the role of diet quality in late-life depression.

Beyond the impact of individual factors, we observed a clear dose–response pattern: the fewer risk factors present, the lower the likelihood of depressive symptoms. Older adults who maintained multiple protective behaviors and health statuses such as non-smoking, adequate sleep, and physical activity had lower odds of depressive symptoms compared with those who accumulated several risk factors. This suggests that the cumulative effect of healthy lifestyle and health status may be greater than the sum of individual effects, highlighting the importance of integral intervention strategies in depression prevention.

Previous studies found lower educational attainment has frequently been associated with greater depressive symptoms, likely mediated by differences in health literacy, income, and access to resources, though the strength of this association may vary by cultural context [2]. Our study also found such association. Lower educational levels may increase vulnerability to depression through limited understanding of health information, barriers to healthcare access, and fewer economic or social resources to buffer stress. Furthermore, education influences lifelong employment opportunities and socioeconomic stability, both of which are well-documented social determinants of mental health. These findings highlight the need to explore social determinants of health more comprehensively and reinforce the importance of addressing social inequities when designing strategies for late-life depression prevention.

The strengths of this study include its comprehensive examination of multiple factors associated with depressive symptoms, considering general health, functional, and modifiable lifestyle factors simultaneously. By evaluating these factors both individually and jointly, the study was able to account for potential intercorrelations and provide a more holistic understanding of their associations with depressive symptoms. There are several noteworthy limitations of the study. First, depressive symptoms were assessed using the self-reported PHQ-9, which is subject to recall bias, social desirability, and potential misclassification of depressive symptoms. Second, perceived health status and subjective cognitive functioning were measured using single-item questions that lack psychometric validation. As a result, these measures may not fully capture the intended constructs, potentially introducing measurement error and weakening the strength of related inferences. Another key limitation is the cross-sectional nature of the study, which precludes the establishment of causal relationships between these factors and depression. The findings therefore indicate associations rather than causality. Additionally, the use of complete-case analysis may introduce bias if the missing data are not missing at random. Although comparisons between participants with complete data and those excluded due to missing data indicated comparable demographic and health characteristics, suggesting that the risk of bias is limited, the exclusion of individuals with missing data may still reduce generalizability. Finally, several well-documented determinants of late-life depression, such as social and relational context, major life events, and psychiatric history (including prior depression), were not included in our analysis because these variables are not collected in NHANES. As a result, the model does not capture the full spectrum of factors influencing depression in older adults, and the findings should be interpreted as reflecting associations among the variables available in the dataset rather than the complete set of determinants.

## 5. Conclusions

In summary, this study highlights the multifaceted nature of depression risk factors in older adults, including physical health, lifestyle behaviors, and psychosocial well-being. The findings underscore the potential importance of integrated health promotion strategies that simultaneously target these factors. Evidence-based interventions should be designed to promote healthy weight, blood pressure control, adequate sleep, smoking cessation, moderate alcohol use, and regular physical activity, with consideration of participants’ psychosocial context. Targeted programs in community or clinical settings may prioritize individuals with multiple risk factors to maximize impact. Future longitudinal observational and interventional studies are needed to confirm the observed associations, clarify the direction of causality, and determine which combinations and intensities of health behaviors and risk-factor modifications are most effective in reducing depression. By targeting modifiable factors, this study provides a foundation for developing holistic interventions aimed at reducing the burden of depression among older adults.

## Figures and Tables

**Table 1 geriatrics-11-00005-t001:** Sociodemographic Characteristics of Study Participants (Weighted Mean or %) (n = 3942).

Characteristics	Overall	Presence of Depressive Symptoms (PHQ ≥ 10) (7.3%)	No Depressive Symptoms (92.7%)	*p*-Value
**Age** (years): Mean (SD)	72.69 (4.96)	72.47 (5.32)	72.71 (4.93)	0.611
**Gender** (%)				0.013
Male	44.5%	33.4%	45.4%	
Female	55.5%	66.6%	54.6%	
**Race/Ethnicity** (%)				0.034
White	78.7%	74.0%	79.1%	
Black	8.0%	8.7%	7.9%	
Hispanic	7.3%	12.1%	6.9%	
Others/Multiracial	6.0%	5.2%	6.1%	
**Marital Status** (%)				0.086
Married/Living with partner	61.3%	52.0%	62.0%	
Widowed/Divorced/Separated	35.4%	44.5%	34.6%	
Never married	3.4%	3.6%	3.4%	
**Education** (%)				<0.001
Less than high school	15.0%	29.5%	13.8%	
High School	24.0%	24.3%	24.0%	
Some college/AA degree	30.2%	33.3%	30.0%	
College or above	30.8%	12.9%	32.2%	

**Table 2 geriatrics-11-00005-t002:** General Health, Functional Status, and Lifestyle Factors by Presence of Depressive Symptoms (Weighted).

	Overall	Presence of Depressive Symptoms	No Depressive Symptoms	*p*-Value
**Perceived Health Condition**				<0.001
Very good or excellent	40.9%	7.9%	43.5%	
Good	38.9%	30.8%	39.5%	
Fair or Poor	20.2%	61.3%	17.0%	
**Physical Functioning**				<0.001
No physical difficulty	91.8%	61.9%	93.7%	
Having physical difficulties	8.2%	32.1%	6.3%	
**Cognitive Issue Experience**				<0.001
No experience	88.8%	63.6%	90.8%	
Confusion/Memory problems	11.2%	36.4%	9.2%	
**BMI**				<0.001
Underweight	0.9%	0.7%	0.9%	
Normal weight	22.6%	21.7%	22.6%	
Overweight	37.1%	23.1%	38.2%	
Obesity	39.4%	54.5%	38.2%	
**Hypertension**				0.016
Non-hypertensive	28.2%	17.3%	29.0%	
Hypertensive	71.8%	82.7%	71.0%	
**Sleep**				0.040
Recommended	68.0%	48.5%	69.6%	
Short	14.8%	21.1%	14.3%	
Long	17.2%	30.4%	16.1%	
**Smoking Status**				0.009
Nonsmoker	50.3%	41.2%	51.0%	
Former smoker	41.0%	43.9%	40.8%	
Current Smoker	8.7%	14.9%	8.2%	
**Physical Activity**				0.039
Meet the Guideline	50.6%	41.2%	51.3%	
Does not meet the Guideline	49.4%	58.8%	48.7%	
**Past-year Alcohol Drink**				<0.001
Never Drinking	39.6%	51.9%	38.7%	
Occasionally Drinking	34.0%	35.0%	33.9%	
Regularly Drinking	26.3%	13.0%	27.4%	
**Dietary quality index:** mean (SD)	54.72 (12.64)	53.75 (12.78)	54.80 (12.62)	0.331

**Table 3 geriatrics-11-00005-t003:** Logistic Regression of Factors Associated with Depressive symptoms (Weighted).

Factors	AOR	95% CI	*p*-Value
**Perceived Health Condition**			
Very good or excellent	reference		
Good	2.09	0.71–6.21	0.168
Fair or Poor	5.90	2.68–12.97	<0.001 ***
**Physical Functioning**			
No physical difficulty	reference		
Having physical difficulties	2.81	1.08–7.35	0.036 *
**Cognitive Issue Experience**			
No experience	reference		
Confusion/Memory problems	1.61	0.71–3.62	0.233
**BMI**			
Normal weight	reference		
Underweight	1.72	0.54–5.45	0.332
Overweight	0.29	0.09–0.95	0.042 *
Obesity	0.62	0.29–1.33	0.201
**Hypertension**			
Non-hypertensive	reference		
Hypertensive	3.08	1.08–8.82	0.038 *
**Sleep**			
Recommended	reference		
Short	1.91	0.62–5.85	0.236
Long	2.25	1.19–4.26	0.017 *
**Smoking**			
Never Smoking	reference		
Former Smoker	1.20	0.63–2.31	0.551
Current Smoker	1.19	0.46–3.07	0.701
**Physical Activity**			
Meet the Guideline	reference		
Does not meet the Guideline	0.89	0.52–1.53	0.664
**Past-year Alcohol Drink**			
Never Drinking	reference		
Occasionally Drinking	1.17	0.64–2.13	0.598
Regularly Drinking	0.45	0.15–1.35	0.142
**Dietary quality index**	1.01	0.98–1.04	0.448

Noted: Adjusted for age, gender, race/ethnicity, marital status, and education. * *p* < 0.05, *** *p* < 0.001.

**Table 4 geriatrics-11-00005-t004:** Association between Number of Risk Factors and Presence of Depressive Symptoms (Weighted).

Number of Risk Factors	AOR	95% CI	*p*-Value
0–1	Reference		
2	1.06	0.26–4.39	0.936
3	1.69	0.47–6.03	0.412
4	5.04	1.35–18.89	0.017 *
5	5.76	1.48–22.48	0.013 *
6	20.94	5.56–78.81	<0.001 ***
7–9	27.89	8.46–91.95	<0.001 ***

Noted: Adjusted for age, gender, race/ethnicity, marital status, and education. * *p* < 0.05, *** *p* < 0.001.

## Data Availability

The original data presented in the study are openly available in NHANES at https://wwwn.cdc.gov/nchs/nhanes/default.aspx. Accessed on 5 June 2025.

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
