# Peer review of "Individual and Cumulative Health and Lifestyle Risk Factors for Depressive Symptoms in Older Adults: Evidence from NHANES"

_geriatrics, 2026, doi:10.3390/geriatrics11010005_

Round 1

Reviewer 1 Report

Comments and Suggestions for Authors

Dear authors,

Below are some key recommendations that could significantly strengthen this work and enhance its scientific impact.

Introduction

Gap in the combined analysis of factors: Although previous studies have investigated individual determinants associated with depression in older adults, few have examined how these factors interact or accumulate to influence depressive symptoms.

Lack of a multifactorial approach: The existing literature does not adequately capture the interdependent and multifaceted nature of depression in the elderly population.

Methods

Variable limitations: Data from later NHANES cycles were excluded due to the absence of key variables such as dietary quality and functional status, thereby restricting the comprehensiveness of the analysis.

Cross-sectional design: The study employs a cross-sectional design, which prevents the establishment of causal relationships between the analysed factors and depression.

Exclusion of participants with incomplete data: The use of complete-case analysis may introduce bias by excluding individuals with missing information.

Results

Lack of significant differences in some factors: No significant differences were observed between participants with and without depression regarding dietary quality, which may reflect limitations in the sensitivity of the measures employed.

Limited impact of certain factors: Some variables, such as self-reported cognitive difficulties and smoking status, did not show statistically significant associations with depression, suggesting the need for further investigation.

Discussion

Causality not established: The cross-sectional nature of the study precludes conclusions regarding the causal direction between the examined factors and depression.

Complexity in the relationship with alcohol consumption: The observed association between regular alcohol intake and a lower risk of depression is complex and warrants deeper investigation to avoid misinterpretation.

Influence of social determinants: The relationship between lower educational attainment and higher prevalence of depression highlights the need to explore social determinants of health more comprehensively.

Conclusion

Need for longitudinal studies: Future research should adopt longitudinal designs to confirm the observed associations and clarify the direction of causality.

Lack of specificity in interventions: Although the study advocates for integrated health promotion strategies, it does not specify which combinations or intensities of interventions would be most effective in preventing depression among older adults.

Author Response

Introduction

[Comment 1] Gap in the combined analysis of factors: Although previous studies have investigated individual determinants associated with depression in older adults, few have examined how these factors interact or accumulate to influence depressive symptoms.

[Response 1] We stated in the Introduction that although these factors have been examined individually, few studies have investigated how they “in combination” influence depression among older adults. For clarification, we have changed “both individually and in combination” to “the accumulation of these factors”.

[Comment 2] Lack of a multifactorial approach: The existing literature does not adequately capture the interdependent and multifaceted nature of depression in the elderly population.

[Response 2] Thank you for this observation. We have added literature related to the interdependent and multifaceted nature of depression in older adults to the Introduction section, to better contextualize our study within the existing research.

Methods

[Comment 3] Variable limitations: Data from later NHANES cycles were excluded due to the absence of key variables such as dietary quality and functional status, thereby restricting the comprehensiveness of the analysis.

[Response 3] Thank you for this comment. We agree that more recent NHANES cycles would have been valuable; however, key variables needed for our analysis, specifically dietary quality and functional status, were not consistently collected in later cycles. To maintain methodological consistency, we limited the analysis to the 2013–2018 cycles where all required variables were available. Importantly, the combined three cycles provided an adequate sample size (N = 3,942) to ensure sufficient statistical power for our analyses. Therefore, although the inclusion of later cycles was not feasible, the sample retained is robust and representative.

[Comment 4] Cross-sectional design: The study employs a cross-sectional design, which prevents the establishment of causal relationships between the analysed factors and depression.

[Response 4] We indicated in the Discussion that a key limitation of this study is the cross-sectional design, “which precludes the establishment of causal relationships between these factors and depression. The findings therefore indicate associations rather than causality” .

[Comment 5] Exclusion of participants with incomplete data: The use of complete-case analysis may introduce bias by excluding individuals with missing information.

[Response 5] Thank you for raising this important point. We acknowledge that complete-case analysis may introduce bias if the missing data are not completely at random. However, we chose this approach to preserve the internal consistency of the dataset by ensuring that all included participants have valid information for all variables in the models.

To assess the potential impact of missing data, we compared key demographic and health characteristics between participants with complete data and those excluded due to missing information. These differences were minimal and not significant, suggesting that the risk of bias is limited. We have added this limitation to the Discussion to acknowledge the possible bias from missing data and lack of the generalizability of the results.

Results

[Comment 6] Lack of significant differences in some factors: No significant differences were observed between participants with and without depression regarding dietary quality, which may reflect limitations in the sensitivity of the measures employed.

[Response 6] Thank you for raising this important point. We have added this to the Discussion to acknowledge that this discrepancy may reflect age-related differences in dietary patterns or increased measurement error in dietary recall among older adults.

[Comment 7]  Limited impact of certain factors: Some variables, such as self-reported cognitive difficulties and smoking status, did not show statistically significant associations with depression, suggesting the need for further investigation.

[Response 7] Cognitive impairment and smoking were strongly associated with depression in univariate analysis, but these associations were no longer statistically significant in multivariable models. This may reflect collinearity with other factors, such as physical functional status and chronic disease. Nevertheless, the elevated ORs suggest a potentially meaningful association that warrants further investigation. We have added this to the Discussion.

Discussion

[Comment 8] Causality not established: The cross-sectional nature of the study precludes conclusions regarding the causal direction between the examined factors and depression.

[Response 8] We indicated in the Discussion that a key limitation of this study is the cross-sectional design, “which precludes the establishment of causal relationships between these factors and depression. The findings therefore indicate associations rather than causality”  

[Comment 9] Complexity in the relationship with alcohol consumption: The observed association between regular alcohol intake and a lower risk of depression is complex and warrants deeper investigation to avoid misinterpretation.

[Response 9] We indicated in the Discussion that the relationship between alcohol use and mental health is multifaceted, highlighting the need for cautious interpretation and careful public health messaging. Per the reviewer’s comment, we have added that further investigation is needed to avoid misinterpretation.

[Comment 10] Influence of social determinants: The relationship between lower educational attainment and higher prevalence of depression highlights the need to explore social determinants of health more comprehensively.

[Response 10] We discussed that the observed association between lower education and depression reinforces the importance of addressing social inequities when designing strategies for the prevention of late-life depression. Per the reviewer’s suggestion, we also added that these findings highlight the need to explore social determinants of health more comprehensively.

Conclusion

[Comment 11] Need for longitudinal studies: Future research should adopt longitudinal designs to confirm the observed associations and clarify the direction of causality.

[Response 11] Based on your comment, we have revised the Conclusion section to note that longitudinal studies are needed to confirm the observed associations and clarify the direction of causality.

[Comment 12] Lack of specificity in interventions: Although the study advocates for integrated health promotion strategies, it does not specify which combinations or intensities of interventions would be most effective in preventing depression among older adults.

[Response 12] Thank you for this insightful comment. We agree that the study does not identify the specific combinations or intensities of interventions that would be most effective in preventing depression among older adults. Our analysis was based on observational cross-sectional data, which allows us to identify associations between individual health factors and depressive symptoms but does not permit evaluation of intervention packages or optimal intervention intensity. We have clarified this in the Conclusion by noting that while our findings support the value of integrated health promotion strategies, future intervention studies are needed to determine which combinations and intensities of health behaviors and risk-factor modifications are most effective in reducing depression among older adults.

Reviewer 2 Report

Comments and Suggestions for Authors

The manuscript “Individual and Cumulative Health and Lifestyle Risk Factors for 2 Depression in Older Adults: Evidence from NHANES” addresses an important and underexplored topic — the multifactorial determinants of late-life depression. The use of NHANES data and the attempt to analyze both individual and cumulative risk factors are valuable contributions. For these reasons, I believe the topic might suit the scope of Geriatrics.

Notwithstanding, I believe the manuscript has many shortcomings and would benefit from working on several topics. As suggestions, I would like to point out some possible aspects that I believe should be worked, and which I will detail next.

  • The central variable of this study—depression—is assessed via self-report (PHQ-9). While the PHQ-9 is widely validated, self-report measures are subject to recall bias, social desirability effects, and under- or overreporting of symptoms. These limitations should be explicitly acknowledged, as they directly affect the robustness of the findings.

  • Several other critical variables, such as perceived health status and subjective cognitive functioning, are measured through single-item indicators. The lack of multi-item validated scales raises concerns about reliability and validity. Without known psychometric properties, these measures may not fully capture the constructs they are intended to represent, which weakens the strength of the inferences drawn.

  • The authors state that data from three NHANES cycles over six years were used. This raises the question of whether a longitudinal design could have been pursued. If longitudinal analyses were not feasible, the rationale for opting for a purely cross-sectional approach should be made explicit. As it stands, the cross-sectional nature of the study severely limits causal interpretations, yet this is not sufficiently problematized in the manuscript.

  • The manuscript mentions “65 years and above who were well cognizant of answering the PHQ-9.” This phrasing suggests that some participants may have completed the instrument multiple times across NHANES cycles. If so, potential practice or learning effects need to be considered. Repeated exposure to the same instrument may influence responses and compromise the assumption of independent measurement.

  • The authors frame their approach as comprehensive and holistic. However, the model omits several well-documented determinants of late-life depression, including social and relational context, social support and participation, significant life events (e.g., widowhood), and psychiatric history (including prior depression). The absence of these critical dimensions undermines the claim of comprehensiveness and should be acknowledged as a significant limitation.

  • The limitations section is underdeveloped and does not adequately address the methodological and conceptual issues outlined above. A more critical and reflective discussion is necessary, including consideration of implications for interpretation, alternative explanations, and directions for future research.

  • The conclusion highlights the importance of health promotion strategies, but the discussion is vague and does not meaningfully translate the findings into theoretical or practical implications. The authors should expand this section to address how specific, evidence-based interventions could be designed, targeted, or implemented in light of the study results. Without such elaboration, the paper risks being descriptive rather than actionable.

Good luck with your submission!

Author Response

[Comment 1] The central variable of this study—depression—is assessed via self-report (PHQ-9). While the PHQ-9 is widely validated, self-report measures are subject to recall bias, social desirability effects, and under- or overreporting of symptoms. These limitations should be explicitly acknowledged, as they directly affect the robustness of the findings.

[Response 1] Thank you for this valuable comment. We agree that relying on a self-reported measure such as the PHQ-9 may introduce potential biases, including recall bias, social desirability, and possible under- or overreporting of depressive symptoms. Although the PHQ-9 has demonstrated strong reliability and validity across diverse populations, these limitations are inherent to self-reported mental health assessments. We have now added this limitation to the Discussion section to explicitly acknowledge these issues.

[Comment 2]  Several other critical variables, such as perceived health status and subjective cognitive functioning, are measured through single-item indicators. The lack of multi-item validated scales raises concerns about reliability and validity. Without known psychometric properties, these measures may not fully capture the constructs they are intended to represent, which weakens the strength of the inferences drawn.

[Response 2] Thank you for this insightful comment. We agree that the use of single-item measures for perceived health status and subjective cognitive functioning may limit reliability and validity. These measures are used in NHANES due to feasibility and respondent burden considerations; however, they may not capture the full complexity of these constructs. We have added this limitation to the Discussion and clarified that the lack of multi-item validated scales may weaken the robustness of inferences related to these variables.

[Comment 3] The authors state that data from three NHANES cycles over six years were used. This raises the question of whether a longitudinal design could have been pursued. If longitudinal analyses were not feasible, the rationale for opting for a purely cross-sectional approach should be made explicit. As it stands, the cross-sectional nature of the study severely limits causal interpretations, yet this is not sufficiently problematized in the manuscript.

[Response 3] We agree that longitudinal data would provide stronger evidence for causal relationships. However, the NHANES data used in this study are collected as repeated cross-sectional cycles, and participants are not followed over time; therefore, a longitudinal analysis at the individual level is not feasible. We have clarified in the Methods and Discussion sections that the study is cross-sectional by design and emphasized that the findings reflect associations rather than causality. We also highlighted this limitation to ensure cautious interpretation of the results.

[Comment 4]  The manuscript mentions “65 years and above who were well cognizant of answering the PHQ-9.” This phrasing suggests that some participants may have completed the instrument multiple times across NHANES cycles. If so, potential practice or learning effects need to be considered. Repeated exposure to the same instrument may influence responses and compromise the assumption of independent measurement.

[Response 4] We would like to clarify that NHANES is a repeated cross-sectional survey, and participants are not followed across cycles. Each individual is surveyed only once, so there is no repeated exposure to the PHQ-9 for the same participant. Therefore, concerns about practice or learning effects and violations of independence do not apply to our analysis. We have clarified this point in the Methods section to avoid any misunderstanding.

[Comment 5] The authors frame their approach as comprehensive and holistic. However, the model omits several well-documented determinants of late-life depression, including social and relational context, social support and participation, significant life events (e.g., widowhood), and psychiatric history (including prior depression). The absence of these critical dimensions undermines the claim of comprehensiveness and should be acknowledged as a significant limitation.

[Response 5] Thank you for this insightful comment. We agree that our study did not include several well-documented determinants of late-life depression, such as social and relational context, social support and participation, major life events (e.g., widowhood), and psychiatric history (including prior depression). These factors were not included because they are not collected in the NHANES dataset. While our analysis focused on physical health, lifestyle behaviors, and selected psychosocial factors available in NHANES, we recognize that omitting these variables limits the comprehensiveness of the model. We have explicitly acknowledged this as a significant limitation in the manuscript, emphasizing that the findings reflect associations among the included factors and do not capture the full spectrum of determinants of late-life depression.

[Comment 6] The limitations section is underdeveloped and does not adequately address the methodological and conceptual issues outlined above. A more critical and reflective discussion is necessary, including consideration of implications for interpretation, alternative explanations, and directions for future research.

[Response 6] Thank you for this important feedback. We agree that the Limitations section needed a more critical and reflective discussion. We have substantially revised it to explicitly address key methodological and conceptual issues, including the cross-sectional design, reliance on self-reported measures (e.g., PHQ-9), use of single-item indicators for certain constructs, and the exclusion of important determinants of late-life depression not captured in NHANES (e.g., social support, major life events, psychiatric history). We have also highlighted how these limitations affect interpretation, considered alternative explanations for the findings, and suggested directions for future longitudinal and interventional research to better understand causal relationships and the comprehensive set of factors influencing depression in older adults.

[Comment 7]  The conclusion highlights the importance of health promotion strategies, but the discussion is vague and does not meaningfully translate the findings into theoretical or practical implications. The authors should expand this section to address how specific, evidence-based interventions could be designed, targeted, or implemented in light of the study results. Without such elaboration, the paper risks being descriptive rather than actionable.

[Response 7] Thank you for this important comment. We agree that the original Conclusion could be strengthened by explicitly translating the findings into theoretical and practical implications. We have revised the section to discuss how the study’s results could inform the design, targeting, and implementation of evidence-based health promotion interventions for older adults. Specifically, we emphasize targeting interventions to modifiable risk factors such as physical health, lifestyle behaviors, and psychosocial well-being, and suggest that combined approaches addressing multiple factors simultaneously may be most effective. We also highlight the need for longitudinal and intervention studies to determine optimal combinations and intensities of these strategies. 

Reviewer 3 Report

Comments and Suggestions for Authors

Using data from the National Health and Nutrition Examination Survey (NHANES), this study evaluated associations of general health/chronic conditions, physical and daily functioning, and lifestyle behaviors with depressive symptoms in individuals 65 years of age and older. Multivariable logistic regression models were used to evaluate health, functioning and lifestyle factors predicting depression groups (yes/no) both individually and combined.  Participants with increased health risk factors (6 or more) compared to those with only one or no risks were significantly more likely to report increased depressive symptoms.

Strong paper. I have only minor suggestions. But consider using depressive symptoms throughout the manuscript, as clinical depression is not being diagnosed.

  1. 189: what does SGA refer to?
  2. 254: Table 3. I suggest using only two places past the decimal for AOR and Confidence intervals.
  3. 257: Table 4. I suggest using only two places past the decimal for AOR and Confidence intervals.
  4. 261: “the elderly population” this is pejorative. Could just say “adult population aged 65 and above….:

Author Response

[Comment 1] Strong paper. I have only minor suggestions. But consider using depressive symptoms throughout the manuscript, as clinical depression is not being diagnosed.

[Response 1] Thank you for this comment. We acknowledge that the PHQ-9 score is a measure of depressive symptoms rather than a clinical diagnosis of major depression. Although we used the standard cutoff (scores ≥10) to categorize participants for interpretability, we agree that “depressive symptoms” is a more precise term. We have revised the manuscript accordingly to use “depressive symptoms” throughout.

[Comment 2] 189: what does SGA refer to?

[Response 2] We apologize for the typo. It should be DGA (Dietary Guidelines for Americans). We have corrected this error.

[Comment 3]  254: Table 3. I suggest using only two places past the decimal for AOR and Confidence intervals.

[Response 3] Thank you for the suggestion. We have revised the manuscript to report AOR and 95% confidence intervals with two decimal places throughout the text and tables.

[Comment 4]  257: Table 4. I suggest using only two places past the decimal for AOR and Confidence intervals.

[Response 4] Thank you for the suggestion. We have made the revisions.

[Comment 5] 261: “the elderly population” this is pejorative. Could just say “adult population aged 65 and above….:

[Response 5] Thank you for this suggestion. We have revised the wording throughout the manuscript to refer to “adults aged 65 and above” instead of “the elderly population” to ensure neutral and respectful language.

Round 2

Reviewer 1 Report

Comments and Suggestions for Authors

Dear authors,

The article provides a comprehensive analysis of individual and cumulative risk factors for depressive symptoms in older adults using data from the NHANES.

However, several scientific limitations can be identified:

Cross-sectional design:
The study relies on cross-sectional data, which restricts the ability to infer causal relationships between risk factors and depressive symptoms. Longitudinal studies are required to confirm the directionality of the observed associations.

Self-reported measures:
Key variables including perceived health status and cognitive functioning were assessed through self-report. This approach may introduce recall bias and reduce the accuracy of the resulting inferences.

Lack of contextual variables:
The study omits important social and contextual determinants, such as social support, major life events, and psychiatric history, all of which are known to influence depression in older populations.

Diet quality:
Although previous research has demonstrated an association between the Healthy Eating Index (HEI-2015) and depression, the present study found no significant relationship.

This discrepancy may reflect age-related differences in dietary patterns or measurement error inherent to dietary recall methods.

Future research should investigate this association using longitudinal designs and alternative HEI classifications.

Collinearity among variables:
Some associations that were significant in univariate analyses such as that between cognitive problems and depressive symptoms were no longer significant in multivariable models.

This pattern may suggest collinearity among variables such as functional status and chronic conditions, warranting further methodological scrutiny.

Exclusion of missing data:
The complete-case analysis may introduce bias if data are not missing at random. Although the authors state that excluded participants were demographically and clinically like those included, such exclusions may still constrain the generalizability of the findings.

Lifestyle factors:
The reported association between alcohol consumption and depressive symptoms is described as complex, yet the analysis does not sufficiently distinguish between the effects of moderate versus excessive consumption.

Evidence-based interventions:
While the study advocates integrated health promotion strategies, it does not elaborate on which specific interventions may be most effective or how these could be adapted to different settings.

Taken together, these limitations highlight important avenues for future research, particularly longitudinal and interdisciplinary studies that incorporate a broader range of health determinants and employ more robust measures of risk factors.

Reviewer 2 Report

Comments and Suggestions for Authors

Dear authors, 

I believe you have addressed all my previous concerns with the manuscript. Good luck with your publication!